# Humoral response after SARS-CoV-2 mRNA vaccines in dialysis patients: Integrating anti-SARS-CoV-2 Spike-Protein-RBD antibody monitoring to manage dialysis centers in pandemic times

**Thomas Bachelet**[ID]\*, **Jean-Philippe Bourdenx, Charlie Martinez, Simon Mucha, Philippe Martin-Dupont, Valerie Perier, Antoine Pommereau**

Clinique Saint-Augustin-CTMR, ELSAN, Bordeaux, France

\* thomas.bachelet@mailo.com

## Abstract

Dialysis patients are both the most likely to benefit from vaccine protection against SARS-CoV-2 and at the highest risk of not developing an immune response. Data from the medical field are thus mandatory. We report our experience with a BNT162b2-mRNA vaccine in a retrospective analysis of 241 dialysis patients including 193 who underwent anti-Spike-Protein-Receptor-Binding-Domain (RBD) IgG analysis. We show that a pro-active vaccine campaign is effective in convincing most patients to be vaccinated (95%) and frequently elicits a specific antibody response (94.3% after two doses and 98.4% after three doses). Only immunocompromised Status is associated with lack of seroconversion (OR 7.6 [1.5–38.2], p = 0.02). We also identify factors associated with low response (last quartile; IgG<500AU/mL): immunocompromised status, age, absence of RAAS inhibitors, low lymphocytes count, high C Reactive Protein; and with high response (high quartile; IgG>7000AU/mL): age ; previous SARS-CoV-2 infection and active Cancer. From this experience, we propose a strategy integrating anti-spike IgG monitoring to guide revaccination and dialysis center management in pandemic times.

## Introduction

Patients on maintenance dialysis are at risk of developing severe caused by SARS-CoV-2 [1]. There is an increased risk of hospitalization in this group, and a reported 10 to 20% of patients with SARS-CoV-2 infection have died in hemodialysis centers in France since the start of pandemic [2]. Thus, these high-risk patients were given priority in international vaccination campaigns [3–6]. Conversely, several decades of experience with the hepatitis B virus vaccine in this group have shown that these patients have an altered immune response and decreased vaccine response [7]. Dialysis patients are therefore both the most likely to benefit from vaccine protection and at the highest risk of not developing an immune response.

**Data Availability Statement:** All relevant data are within the manuscript and its Supporting Information files.

**Funding:** The author(s) received no specific funding for this work.

**Competing interests:** The authors have declared that no competing interests exist.

**Abbreviations:** SARS-CoV-2, Severe Acute Respiratory Syndrome Coronavirus 2; COVID-19, Coronavirus disease 2019; iRAAS, inhibitors of Renin Angiotensin Aldosteron System; RBD, Receptor Binding Domain.

Several vaccines have been approved for SARS-CoV-2 infection, in particular, new generation lipid nanoparticle delivery of antigen encoding mRNA vaccines BNT162b2 (Pfizer-BioNTech®) [8] and mRNA-1273 (Moderna®) [9]. These vaccines have been shown to have a high level of protection which has been confirmed in a nationwide mass vaccination setting [10]. They also elicit an antibody response [11, 12] that seems to persist over the time [13]. One aspect of the immune response can be easily characterized and quantified in routine practice using the recent test on seroconversion of IgG antibodies against the receptor binding domain protein of the S1 subunit of the spike protein of SARS-CoV-2 (Spike-Protein-RBD IgG). The humoral response against SARS-CoV-2 in dialysis patients is the subject of intense research [14]. Seminal studies have confirmed the efficacy of the new mRNA vaccines in dialysis patients for seroconversion of the Spike-Protein-RBD IgG [15, 16]. On the other hand, there is concern about transplanted patients receiving maintenance immunosuppression [17, 18].

We report our experience with vaccines BNT162b2 (Pfizer-BioNTech) in a retrospective observational single center study of dialysis patients and proposed to integrate anti-SARS-CoV-2 Spike-Protein-RBD antibody monitoring in the management of dialysis centers facing the SARS-CoV2 pandemic.

## Material and methods

### Study design

Retrospective single center observational cohort study according to modified STROBE statements.

### Setting and participants

As of January 2021, dialysis patients were given access to a local vaccination campaign according to French recommendations (https://www.has-sante.fr/jcms/p_3234097/fr/modification-du-schema-vaccinal-contre-le-sars-cov-2-dans-le-nouveau-contexte-epidemique). Dialysis patients were recruited to receive a two injection-scheme of the BNT162b2 mRNA vaccine. Resident patients in nursing homes were vaccinated in their institute. Previous SARS-CoV-2 -infected patients received only one injection. Home dialysis patients returned to the center to be vaccinated. Individual information on the balance benefit risk of the vaccine was provided to each patient by the medical staff.

### Variables: Detection and characterization of SARS-CoV-2 antibodies

Patients were tested by the SARS-CoV-2 IgG Architect system (targeting the nucleocapsid antigen) on the day of the first injection (Abbott©). Patients were tested with SARS-CoV-2 IgG serologic assays targeting Spike-Protein-RBD one month after the second dose of vaccine. The IgG II Quant (Abbott©) assay was used for the quantitative measurement of the anti-Spike-Protein-RBD IgG antibodies of SARS-CoV-2. A test was considered to be positive if the IgG was > 50 AU/ml.

### Data sources

Blood samples were collected during routine dialysis visits in our medical center CTMR (Centre de Traitement des Maladies Rénales, Clinique Saint-Augustin), ELSAN group. Baseline comorbidities, clinical and biological data (hemoglobin, lymphocyte count, C Reactive Protein and albuminemia) were obtained from our electronic medical database (SINED- Groupe Theradial, Medical Computer Systems). Immunocompromised status was characterized by one of the following factors: former or current organ transplant still requiring immunosuppressive therapy, HIV infection, recent (within 6 months) immunosuppressive therapy, or chemotherapy. All patients were informed using the approved written informed consent form.

Their non-opposition to data collection for research purposes was traced in the medical file. Data collection was declared to the French Commission Nationale de l'Informatique et des Libertés, registration 2222259. This protocol was submitted to the approbation of Elsan Group Institutional Review Board.

## Outcomes

The two main outcomes of the study were the global vaccination rate and the rate of seroconversion to SARS-CoV-2 anti-Spike-Protein-RBD antibodies. The secondary outcomes included the factors associated with seroconversion to SARS-CoV-2 anti-Spike-Protein-RBD antibodies and the low and high quartiles of quantitative IgG response.

## Statistical analysis

The McNemar, Chi-square or Fisher test was used for categorical variables, and Student t test for quantitative variables. Risk factors associated with SARS-CoV-2 anti-Spike-Protein-RBD with p<0.1 on univariate analysis were included in multivariate analysis. Covariates independently associated with the outcome were selected by iterative backward elimination and only those having p < .05 were retained. All analyses were performed with JMP.10 (2012, SAS Institute Inc, Cary, NC, USA).

## Results

### Baseline characteristics

A total of 241 patients were included (mean age 73.8±12.6 years old). The SARS-CoV-2 IgG antinucleocapsid test was performed in 221 patients. Nineteen patients (7.9%) had been infected with SARS-CoV-2 the year before. Twelve out of the available 15 tested patients previously infected with SARS CoV2 were still seropositive (80%). On the opposite, two patients were found to be seropositive for the SARS-CoV-2 -IgG antinucleocapsid but had no symptoms of COVID-19. These last two patients only received one dose of vaccine and showed low titers of SARS-CoV-2 anti-Spike-Protein-RBD antibodies, that were below the positivity threshold. These patients were thus reclassified as false positives and programmed for a second dose. They were not included in the final analysis. Five other patients developed an infection in the days following vaccination. Their second injection was postponed, and they were also not included in the seroconversion analysis. Transplanted or deceased patients during the period between the two vaccine injections were also excluded from the seroconversion analysis as well as home dialysis patients, the two patients contraindicated for a suspicion of infection the day of vaccination which required antibiotherapy or any patients that did not receive the two injections (Fig 1, Flowchart).

### Vaccination rate

The acceptance was high with a SARS-CoV-2 vaccination initiated in 95% of the patient from our cohort (227/241), attesting that patients were aware of the risk of severe SARS-CoV-2 infection and had been convinced by the information communicated to them personally. This vaccination rate was higher than for our 2020 anti-influenza vaccine campaign (78%).

### Seroconversion study

Overall, 193 dialysis patients were included in the SARS-CoV-2 anti-Spike-Protein-RBD seroconversion study. Patient characteristics and comorbidities are described in Table 1. At a median of 32.7±0.5 days after the second dose of vaccine, 182 (94.3%) of the group were seropositive for SARS-CoV-2 anti-Spike IgG. The mean anti- Spike-Protein-RBD titer was 5764

# Flowchart

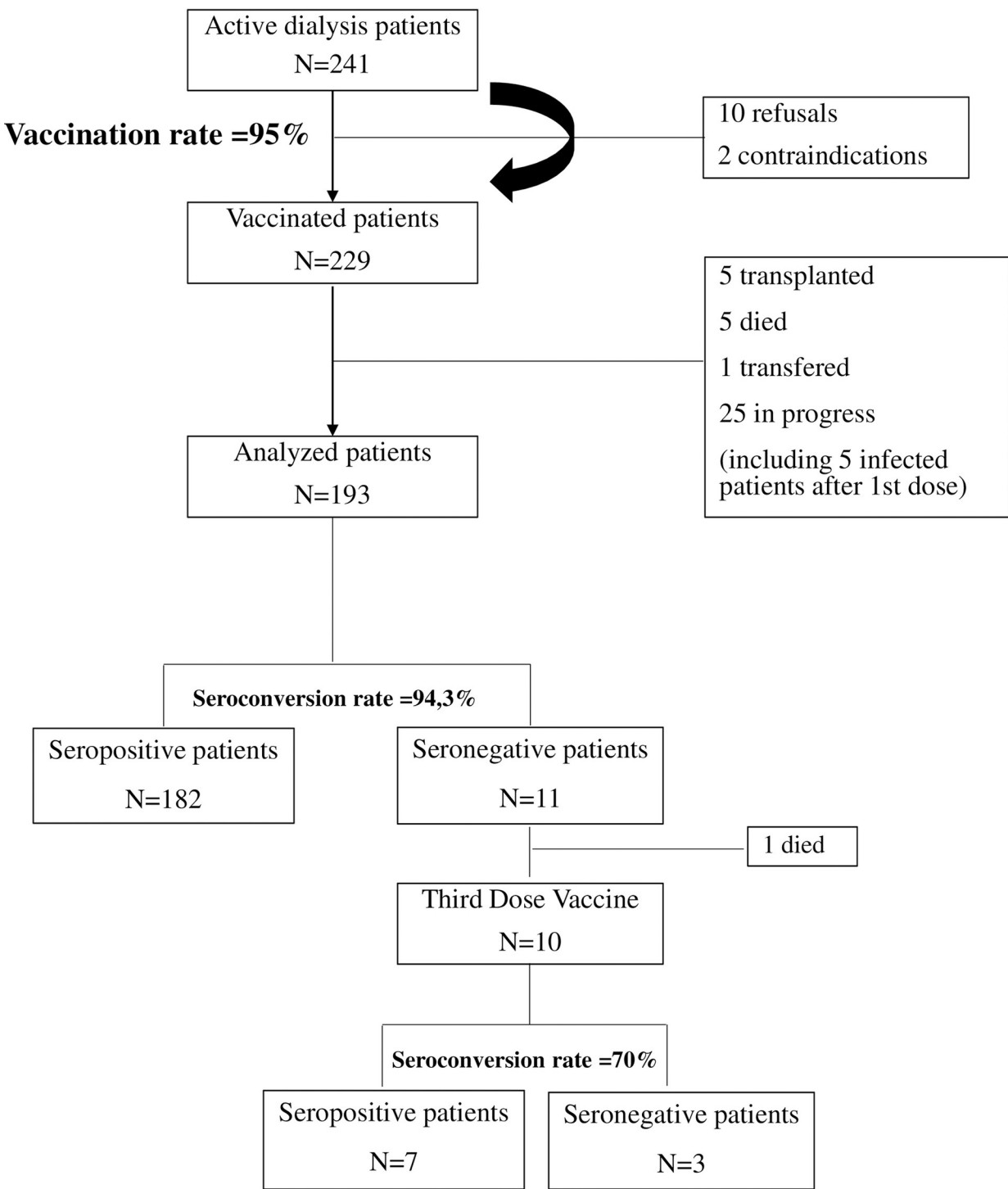

**Global seroconversion rate =98.4%**

Seropositiviy threshold (Abbott©): Spike-Protein-RBD IgG anti-SARS-CoV-2 > 50AU/mL

**Fig 1. Flowchart of patient receiving BNT162b2-mRNA vaccines and analysed for IgG anti-SARS-CoV-2 Spike-Protein-RBD.**

**Table 1. Baseline characteristics according to IgG anti SARS-CoV2 response*.**

| | ALL PATIENTS N = 193 | SEROPOSITIVE IgG+ anti-SARS-CoV2 N = 182 | SERONEGATIVE IgG- anti-SARS-CoV2 N = 11 | p |
|---|---|---|---|---|
| Sex (N, Woman/Man) | 70/123 | 67/119 | 3/8 | 0.39 |
| Age (years, mean±sd) | 73.7±12.7 | 73.5±12.7 | 76.9±12.5 | 0.40 |
| Time after last vaccination injection (days, mean±sd) | 32.7±6.6 | 32.8±6.7 | 30.5±4.9 | 0.25 |
| Body Mass Index (Kg/m2, mean±sd) | 25.8±4.7 | 25.8±4.7 | 25.3±4.2 | 0.74 |
| Primary Kidney Disease (N, %) | | | | 0.15 |
| Vascular Nephropathy | 44 (23%) | 38 (21%) | 6 (55%) | |
| Uropathy/renal reduction | 14 (7%) | 14 (8%) | 0 | |
| Polycystic Renal Disease | 10 (5%) | 10 (6%) | 0 | |
| Chronic Tubulo-Interstitial Nephropathy | 31 (16%) | 29 (16%) | 2 (18%) | |
| Glomerular Disease | 21 (11%) | 21 (11%) | 0 | |
| Diabetic Nephropathy | 50 (26%) | 48 (26%) | 2 (18%) | |
| Undeterminated Nephropathy | 23 (12%) | 22 (12%) | 1 (9%) | |
| Diabetes Mellitus (N,%) | 75 (39%) | 71 (39%) | 4 (36%) | 0.86 |
| Hypertension (N,%) | 161 (83%) | 151 (83%) | 10 (91%) | 0.46 |
| Cardiac Associated Disease (N,%) | 53 (27%) | 49 (27%) | 4 (36%) | 0.51 |
| Peripheral Arterial Disease (N, %) | 47 (24%) | 42 (23%) | 5 (45%) | 0.12 |
| Chronic Heart Failure (N, %) | 31 (16%) | 29 (16%) | 2 (18%) | 0.81 |
| Chronic Obstructive Pulmonary Disease/Asthma (N,%) | 27 (14%) | 26 (14%) | 1 (9%) | 0.68 |
| Cirrhosis (N, %) | 6 (3%) | 5 (3%) | 1 (9%) | 0.34 |
| Active Cancer (N, %) | 19 (10%) | 16 (9%) | 3 (27%) | 0.09 |
| Autoimmune Disease (N, %) | 10 (5%) | 8 (4%) | 2 (18%) | 0.10 |
| Immunocompromised Status (N, %) | 16 (8%) | 11 (6%) | 5 (45%) | **0.001** |
| History of Kidney Transplantation (N, %) | 20 (10%) | 17 (9%) | 3 (27%) | **0.08** |
| Other Organ Transplantation (N, %) | 4 (2%) | 2 (1%) | 2 (18%) | **0.01** |
| Inhibitor of RAAS Therapy | 74 (38%) | 73 (40%) | 1 (9%) | **0.03** |
| Hemoglobin (g/dl, mean±sd) | 11.4±1.1 | 11.4±1.2 | 11.3±0.7 | 0.86 |
| Lymphocytes (giga/mm3, mean±sd) | 1.10±0.44 | 1.11±0.44 | 0.86±0.37 | **0.05** |
| Serum albumin (g/l, mean±sd) | 36.7±3.8 | 36.8±3.7 | 34.7±4.5 | **0.03** |
| C Reactive Protein (mg/l, mean±sd) | 12.8±21 | 12±21 | 22.5±26 | 0.12 |
| Previous SARS CoV2 infection (>3 months) | 19 (10%) | 19 (13%) | 0 | 0.001 |

* SARS-CoV-2 IgG serologic assays targeting Spike Protein Receptor Binding Domain-RBP (ABBOTT®) with a IgG positive threshold at 50 AU/mL.

AU/ml, the median was 2085 AU/ml. We established the lowest and highest quartitles of IgG anti-Spike-Protein-RBD with a titer of IgG <500AU/mL for the lowest (low responder) and >7000AU/mL for the highest (high responder), respectively.

Seronegative patients were more often immunocompromised (45% vs 6%, p = 0.001), had more frequently other functional organ transplants (18% vs 1%, p = 0.01), were less frequently receiving Angiotensin-Aldosteron System (RAAS) inhibitors (9% vs 40%, p = 0.03) had lower lymphocyte count (0.86±0.14 vs 1.11±0.03, p = 0.05) and lower serum albumin (34.7±1.1 vs 36.8±0.3, p = 0.03).

## Risk factors for low and high responders to vaccination

The only factor associated with the absence of seroconversion (IgG titer <50 AU/mL) on multivariate analysis was immunocompromised status (Table 2A, OR 7.6 [1.5–38.2], p = 0.02).

**Table 2. A. Risk factors associated with the absence of seroconversion Spike-Protein-RBD IgG anti-SARS-CoV-2.** B. Risk factors associated with a low Spike-Protein-RBD IgG anti-SARS-CoV-2 (IgG <500AU/mL, first quartile vs other three quartiles). C. Risk factors associated with a high Spike-Protein-RBD IgG anti-SARS-CoV-2 (IgG >7000AU/mL, Last quartile vs other three quartiles).

| Variables | No patients | No events | Univariate analysis | | | Multivariate analysis | | |
|---|---|---|---|---|---|---|---|---|
| | | | OR | CI | p | OR | CI | p |
| **A.** | | | | | | | | |
| Active Cancer | | | | | | | | |
| Yes | 19 | 3 | 3.6 | [0.75–14] | 0.11 | | | |
| No | 174 | 8 | | | | | | |
| Immunocompromised status | | | | | | | | |
| Yes | 16 | 5 | 12.9 | [2.1–34.5] | 0.0005 | 7.6 | [1.5–38.2] | 0.02 |
| No | 177 | 6 | | | | | | |
| iRAAS Therapy | | | | | | | | |
| Yes | 74 | 1 | 0.14 | [0.01–0.79] | 0.02 | | | |
| No | 119 | 10 | | | | | | |
| Lymphocytes (per Giga/mm3) | 193 | 11 | 0.17 | [0.02–1.01] | 0.054 | | | |
| Serum albumin (per g/L) | 193 | 11 | 0.87 | [0.75–1.02] | 0.054 | | | |
| **B.** | | | | | | | | |
| Age (per years) | 193 | 47 | 1.05 | [1.02–1.09] | 0.001 | 1.06 | [1.02–1.1] | 0.006 |
| Active Cancer | | | | | | | | |
| Yes | 19 | 9 | 3.2 | [1.2–8.6] | 0.02 | | | |
| No | 174 | 10 | | | | | | |
| Immunocompromised status | | | | | | | | |
| Yes | 16 | 8 | 3.5 | [1.2–10.2] | 0.02 | 5.55 | [1.14–30.5] | 0.034 |
| No | 177 | 8 | | | | | | |
| iRAAS Therapy | | | | | | | | |
| Yes | 74 | 8 | 0.25 | [0.1–0.54] | 0.003 | 0.27 | [0.09–0.72] | 0.007 |
| No | 119 | 39 | | | | | | |
| Lymphocytes (per Giga/mm3) | 193 | 47 | 0.15 | [0.05–0.41] | <0.001 | 0.22 | [0.07–0.6] | 0.002 |
| Serum albumin (per g/L) | 193 | 47 | 0.89 | [0.81–0.97] | 0.006 | | | |
| CRP (per mg/L) | 193 | 47 | 1.04 | [1.02–1.06] | <0.001 | 1.04 | [1.02–1.06] | <0.001 |
| **C.** | | | | | | | | |
| Age (per years) | 193 | 47 | 0.95 | [0.93–0.98] | 0.001 | 0.95 | [0.93–0.99] | 0.007 |
| Chronic Heart Failure | | | | | | | | |
| Yes | 31 | 3 | 0.28 | [0.06–0.86] | 0.02 | | | |
| No | 162 | 28 | | | | | | |
| Active Cancer | | | | | | | | |
| Yes | 19 | 0 | not quantifiable | | 0.0008 | not quantifiable | | 0.009 |
| No | 174 | 19 | | | | | | |
| iRAAS Therapy | | | | | | | | |
| Yes | 74 | 24 | 2.0 | [1.03–3.92] | 0.04 | | | |
| No | 119 | 50 | | | | | | |
| Serum albumin (per g/L) | 193 | 47 | 1.11 | [1.01–1.22] | 0.03 | | | |
| CRP (per mg/L) | 193 | 47 | 0.98 | [0.96–1.01] | 0.09 | | | |
| Previous SARS-CoV-2 infection | | | | | | | | |
| Yes | 19 | 15 | 16.2 | [5.37–60.4] | <0.001 | 15.1 | [4.79–58.8] | <0.001 |

(*Continued*)

**Table 2.** (Continued)

| Variables | No patients | No events | Univariate analysis | | | Multivariate analysis | | |
|---|---|---|---|---|---|---|---|---|
| | | | OR | CI | p | OR | CI | p |
| No | 174 | 4 | | | | | | |

Definitions: Active cancer (diagnostic of neoplasia without enough delay to permit a theoretical inscription for a transplantation access); CRP (C Reactive Protein) Immunocompromised status (immunosuppressive treatment of haematological disease interfering with immune system); iRAAS (inhibitor of renin angiotensin aldosterone system); RBP (Receptor Binding Protein).

Age, Body Mass Index, Overweight, Sex, Renal initial nephropathy, Diabetes melitus, Hypertension, Cardiac Associated Disease, Peripheral Arterial Disease, Chronic Heart Failure, Chronic Obstructive Pullmonary Disease or Asthma, Cirrhosis, Active Cancer, Autoimmune Disease, Immunocompromised status, Previous Kidney Transplantation, iRAAS, Hemoglobin, Lymphocytes, Serum albumin and C Reactive Protein were tested in logistic regression for each item and reported only if they were significant in univariate analysis with a p>0.2. Covariates with a p <0.2 in univariate analysis were included in a multivariate model, then eliminated iteratively until only those whose association with the event of interest was significant (p <0.05 was retained).

Univariate then multivariate analysis identified age (OR = 1.06 per year [1.02–1.1], p = 0.006), immunocompromised status (OR 5.55 [1.14–30.5], p = 0.034), absence of iRAAS therapy (OR for iRAAS treatment = 0.27 [0.07–0.6], p = 0.002) and C-Reactive-Protein (OR = 1.04 per mg/L, [1.02–1.06], p<0.001) as risk factors of being a low responder (lowest quartile of IgG anti-Spike-Protein-RBD titer <500AU/mL, Table 2B). On the other hand, previous SARS-CoV-2 infection (15.1 [4.79–58.8], p<0.001), lower age (OR = 0.95 per year [0.93–0.99], p = 0.007) and the absence of an active cancer (OR non quantifiable, p = 0.009) were associated with being a high responder (highest quartile of IgG anti-Spike-Protein-RBD titer >7000AU/mL, Table 2C).

## Outcome

No major adverse events were reported. No additional COVID-19 infections have occurred since February 2021. Meanwhile, revaccination with a third dose of BNT162b2 mRNA vaccine, has been recommended for End-Stage-Renal-Disease patients in France. However, we don't forget that SARS-CoV-2 vaccines should be considered as a rare and precious therapy and consequently should be rationalized [19]. That's why we proposed to first give this third dose to our non-responder patients to a two dose-scheme of the BNT162b2 mRNA vaccine. On the 11 patients who don't seroconvert, one died during the follow-up of a cardiac cause. On the 10 remaining patients, 7 developed a humoral response with Spike-Protein-RBD IgG anti-SARS-CoV-2 reaching the positivity threshold of the kit (50AU/mL), although among the 7, only 3 had IgG titers above 500 AU/mL (Fig 1). Taken together, we found an excellent global seroconversion rate of 98.4% in our cohort after a rationalized strategy for the allocation of the third dose.

## Perspectives

The sole analysis of humoral response may underestimate or overestimate the immunogenicity of the vaccine. Noteworthy, it missed the evaluation of cell-mediated immunity which requires a dedicated specific platform which goes beyond the standard of care [20, 21]. However, the well-known correlation between these two systems may support the hypothesis that high IgG titers could reflect a robust coordinated immune response. We made this hypothesis to design an algorithm integrating anti-SARS-CoV-2 Spike-Protein-RBD antibody monitoring to manage dialysis centers and allocation of the third vaccine dose (Fig 2). The cut off of 500 AU/mL was determined as the IgG titers of the low responder quartile. Incidentally it represents a ratio of ten (x10) for the positivity threshold of the kit. Other population studies should be

## Algorithm

**Fig 2. Proposed algorithm integrating anti-SARS-CoV-2 Spike-Protein-RBD antibody monitoring to manage dialysis centers.**

made to confirm this distribution in other cohorts. We here propose to use this cutoff to rationalize the timing of the distribution of the third vaccine. This third dose is delayed until the IgG titers drop below 500AU/mL with monitoring every 3–6 months during pandemic times. This strategy should prolong the protection conferred by vaccination beyond the supposed 8–9 months after the end of complete SARS-CoV-2 vaccination program, without endangering dialysis patients during epidemic waves. The prospective follow-up of our cohort might demonstrate the safety and efficacy of the proposed algorithm.

## Discussion

The risk of SARS-CoV-2 infection in dialysis patients has been a major concern due to their increased risk of severe illness as well as their need for repeated visits to healthcare facilities. This study shows that a pro-active vaccine campaign can successfully reach the majority of these patients (95%).

We also confirmed that the BNT162b2 mRNA vaccine effectively elicits a specific antibody response against the main antigenic target of the virus (spike protein) in the same proportion as that in seminal reports on the protection from infection in real-life studies. In addition, we demonstrated that the adjunction of a third dose makes possible to attain more than 98% of seroconversion. These results confirmed the immunogenicity of these new generation mRNA vaccines in dialysis patients. There is thus a discrepancy with the results reported in kidney transplantation recipients which hardly succeed to seroconvert in 70% of the patients after three doses [22]. Our study effectively confirmed the immunocompromised status as the main factor predisposing patients to an absence of seroconversion and/or a low IgG anti

SARS-CoV-2 Spike-Protein-RBD response. Other classical conditions interfering with the ability to develop a new immune response were also found to be associated with a low response (age, nutritional state, chronic inflammation, lymphocyte count).

We confirmed that a previous SARS-CoV-2 infection had a booster effect on the intensity of antibody response (in high responder). This suggests that an initial encounter with the complete virus helps trigger a secondary broad immune response to the antigen which makes a link between the IgG levels and the not-analyzed cellular response.

Finally, our results also suggest that RAAS inhibitor therapy may be a new factor associated with a better antibody response to SARS-CoV-2. There is probably an indication bias because RAAS inhibitors may have been given to the healthiest patients. However, RAAS inhibitors also increase ACE2 expression [23]. ACE2 is an enzyme that physiogically counters RAAS activation and is found on the cells of numerous tissues. It is also a receptor for SARS-CoV-2 by binding with its spike protein, thus allowing its entry into host cells [24]. Therefore, the possibility that this overexposed target could trigger an amplified loop in the immune response cannot be excluded and is mentioned here for the first time.

This study has several limitations. Firstly, the specific, retrospective, single center design of the study could result in insufficient collected data to enable unsupervised analysis susceptible to identify unexpected patterns of association. Secondly, the immunological approach using an antibody response alone to evaluate the immune vaccine response may be too simplistic in comparison to systems-biology approaches and evaluation of cell-mediated immunity [20, 21, 25–27]. Thirdly, the timing of the decline in the IgG, the IgG titer which correlates with protection in front of a real-life clinical challenge like the extent of clinical protection created by the antibody response remain uncertain, especially in the presence of emerging variants with mutations on SARS-CoV-2 spike protein.

Taken together, these results confirmed the acceptability, the efficacy, and the safety of the mRNA vaccine BNT162b2 in dialysis patients. They suggest that extensive vaccination campaigns as well as collecting data on the factors associated with the intensity of the humoral response should be continued in vulnerable patients. Based on this experience, we believe that anti-SARS-CoV-2 Spike-Protein-RBD antibody monitoring should help to manage dialysis centers.

## Supporting information

**S1 Data.**
(XLSX)

## Acknowledgments

ELSAN group research initiative.

## Author Contributions

**Conceptualization:** Thomas Bachelet, Jean-Philippe Bourdenx, Charlie Martinez, Simon Mucha, Antoine Pommereau.

**Data curation:** Thomas Bachelet, Jean-Philippe Bourdenx, Charlie Martinez, Simon Mucha, Philippe Martin-Dupont, Antoine Pommereau.

**Formal analysis:** Thomas Bachelet.

**Investigation:** Thomas Bachelet, Jean-Philippe Bourdenx, Charlie Martinez, Simon Mucha, Valerie Perier, Antoine Pommereau.

**Methodology:** Thomas Bachelet, Jean-Philippe Bourdenx, Charlie Martinez, Simon Mucha, Valerie Perier, Antoine Pommereau.

**Project administration:** Thomas Bachelet, Valerie Perier.

**Resources:** Thomas Bachelet, Philippe Martin-Dupont, Valerie Perier.

**Software:** Thomas Bachelet.

**Supervision:** Valerie Perier, Antoine Pommereau.

**Validation:** Thomas Bachelet.

**Writing – original draft:** Thomas Bachelet.

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
