## [Decision Letter · Decision Letter 0]

13 Jul 2021

PONE-D-21-16866

Humoral response after anti-SARS-CoV-2 mRNA vaccines in dialysis patients: Integrating anti-SARS-CoV-2 Spike-Protein-RBD antibody monitoring to manage dialysis centers in pandemic times.

PLOS ONE

Dear Dr. BACHELET,

Thank you for submitting your manuscript to PLOS ONE. After careful consideration, we feel that it has merit but does not fully meet PLOS ONE’s publication criteria as it currently stands. Therefore, we invite you to submit a revised version of the manuscript that addresses the points raised during the review process.

Please provide the details about the methods as the reviewers commented.

We look forward to receiving your revised manuscript.

Kind regards,

Etsuro Ito

Academic Editor

PLOS ONE

Journal Requirements:

3.  Please provide additional details regarding participant consent. In the ethics statement in the Methods and online submission information, please ensure that you have specified  what type you obtained (for instance, written or verbal, and if verbal, how it was documented and witnessed)."

4. In the ethics statement in the manuscript and in the online submission form, please provide additional information about the patient records/samples used in your retrospective study, including the source of the medical records/samples analyzed in this work (e.g. hospital, institution or medical center name).

Reviewers' comments:

Reviewer's Responses to Questions

**Comments to the Author**

1. Is the manuscript technically sound, and do the data support the conclusions?

Reviewer #1: Yes

Reviewer #2: Yes

Reviewer #3: Yes

2. Has the statistical analysis been performed appropriately and rigorously? 

Reviewer #1: Yes

Reviewer #2: Yes

Reviewer #3: Yes

3. Have the authors made all data underlying the findings in their manuscript fully available?

Reviewer #1: Yes

Reviewer #2: Yes

Reviewer #3: Yes

4. Is the manuscript presented in an intelligible fashion and written in standard English?

Reviewer #1: Yes

Reviewer #2: Yes

Reviewer #3: Yes

5. Review Comments to the Author

Reviewer #1: The Authors in "Humoral response after anti-SARS-CoV-2 mRNA vaccines in dialysis patients: Integrating anti-SARS-CoV-2 Spike-Protein-RBD antibody monitoring to manage dialysis centers in pandemic times", describe very well the different factors influencing the response to BNT162b2-mRNA vaccine in dialysis patients. I think this paper is suitable to be accepted but with minor revisions:

1. As age seems to be associated to both low and high response to the vaccine, I suggest to add a figure to this paper in which the author could show the correlation Antibody concentration/Age.

2. Page 7, line 148-150: it seems that something has been skipped in this sentence. I recommend that you construct a meaningful sentence.

3. Authors shown several limitation in their study. Accordingly, as the sole analysis of humoral response may underestimate the immunogenicity of the vaccine, it is critical the evaluation of cell-mediated immunity to estimate the response to the vaccine. On this purpose, I suggest to add in your references, two recent papers about this topic: PMID: 34058052, PMID: 34036720

Reviewer #2: Major comments:

1) Why and how was a cut off of 500 AU/ml set to make a decision about additional vaccination? This value seems to me to be quite arbitrary, as absolute values and cut offs differ between commercially available assays. Furthermore, it is still not clear which specific antibody value correlates with protection against (severe) infections. The algorithm and the limitations of this algorithm should be discussed in more detail in the discussion section.

2) On P7 L148-150 something went wrong. Data seems to be missing and the sentence is not completed.

3) I am not convinced that active cancer is really responsible for higher antibody levels after vaccination. Are there any data about BNT162b2 or other vaccines and active cancer? This should be discussed.

4) The authors should also discuss why (based one which data) they stop vaccinating after the second booster dose. E.g. in hepatitis B, a 4th or 5th dose seems to be promising in some patients.

Minor comments:

5) "Patients on maintenance dialysis are at risk of developing severe disease from Coronavirus 2019 (COVID-19), in severe acute respiratory syndrome coronavirus 2 (SARS-CoV-2)." This sentence of the introduction is not correct. The last part should be "caused by SARS-CoV-2" or something like that.

6) Please use the term "pandemic" instead of "pandemy" throughout the manuscript (e.g. P4 L75).

7) Please delete the following sentence: "Statistical analysis was performed using conventional methods."

8) Please write SARS-CoV-2 and COVID-19 consistently instead of "SARS COV2" or "SARS CoV-2" or "COVID19".

9) On P6 L131 "seroconversion" is written incorrectly.

10) Please write "SARS-CoV-2 vaccination" and not "anti-SARS-CoV-2 vaccination" throughout the manuscript.

11) Ad Table 1. What does "Figru" under the section "Time after last vaccination injection"?

12) "In" is misspelled on P10 L188.

13) What were the 2 contraindications for vaccination?

14) Regarding the algorithm: Under the 3rd dose "IgG > or < 50" is stated. Did you mean 500 as after the second vaccination? What does "Adapted to circulating viral variants" exactly mean?

Reviewer #3: This review is on the statistical aspects of the paper.

The statistical analysis part is well-written. I only have 2 minor comments.

1. page 5, line 114, why the p-value threshold is set at 0.15?

2. Table 1. the p-value should use the same number of significant digits.

6. PLOS authors have the option to publish the peer review history of their article (what does this mean?). If published, this will include your full peer review and any attached files.

Reviewer #1: No

Reviewer #2: No

Reviewer #3: No

---

## [Author Response · Author response to Decision Letter 0]

27 Aug 2021

Humoral response after anti-SARS-CoV-2 mRNA vaccines in dialysis patients: Integrating anti-SARS-CoV-2 Spike-Protein-RBD antibody monitoring to manage dialysis centers in pandemic times.

PLOS ONE

Dear Dr. BACHELET,

Thank you for submitting your manuscript to PLOS ONE. After careful consideration, we feel that it has merit but does not fully meet PLOS ONE’s publication criteria as it currently stands. Therefore, we invite you to submit a revised version of the manuscript that addresses the points raised during the review process.

Please provide the details about the methods as the reviewers commented.

We look forward to receiving your revised manuscript.

Kind regards,

Etsuro Ito

Academic Editor

PLOS ONE

RESPONSE TO REVIEWER

Reviewers' comments:

Reviewer's Responses to Questions

Comments to the Author

1. Is the manuscript technically sound, and do the data support the conclusions?

Reviewer #1: Yes

Reviewer #2: Yes

Reviewer #3: Yes

2. Has the statistical analysis been performed appropriately and rigorously?

Reviewer #1: Yes

Reviewer #2: Yes

Reviewer #3: Yes

3. Have the authors made all data underlying the findings in their manuscript fully available?

Reviewer #1: Yes

Reviewer #2: Yes

Reviewer #3: Yes

4. Is the manuscript presented in an intelligible fashion and written in standard English?

Reviewer #1: Yes

Reviewer #2: Yes

Reviewer #3: Yes

5. Review Comments to the Author

Reviewer #1: The Authors in "Humoral response after anti-SARS-CoV-2 mRNA vaccines in dialysis patients: Integrating anti-SARS-CoV-2 Spike-Protein-RBD antibody monitoring to manage dialysis centers in pandemic times", describe very well the different factors influencing the response to BNT162b2-mRNA vaccine in dialysis patients. I think this paper is suitable to be accepted but with minor revisions:

1. As age seems to be associated to both low and high response to the vaccine, I suggest to add a figure to this paper in which the author could show the correlation Antibody concentration/Age.

RESPONSE 1 : Age is one of the strongest predictive factor associated with variability in the response to vaccination (see for example Gustafson, C. E et al.(2020). Influence of immune aging on vaccine responses. J Allergy Clin Immunol, 145(5), 1309-1321). We found indeed the previously-described correlation between age and response to vaccination in our work. However, additional factors (like immunosuppression or previous SARS-CoV-2 infection) make the linearity of the correlation less convincing (even if we choose a logarithmic presentation of the data). We are not sure that adding this figure would be fully relevant. 

Figure : Anti-SARS-CoV-2 Spike-Protein-RBD IgG titers (AU/mL, Abbott©) according to the age ot the dialysis patients 

2. Page 7, line 148-150: it seems that something has been skipped in this sentence. I recommend that you construct a meaningful sentence.

RESPONSE 2 : We corrected the sentence as followed : « Seronegative patients were more often immunocompromised (45% vs 6%, p=0.001), had more frequently other functional organ transplants (18% vs 1%, p=0.01), were less frequently receiving Angiotensin-Aldosteron System (RAAS) inhibitors (9% vs 40%, p=0.03) had lower lymphocyte count (0.86±0.14 vs 1.11±0.03, p=0.05) and lower serum albumin (34.7±1.1 vs 36.8±0.3, p=0.03). »

3. Authors shown several limitation in their study. Accordingly, as the sole analysis of humoral response may underestimate the immunogenicity of the vaccine, it is critical the evaluation of cell-mediated immunity to estimate the response to the vaccine. On this purpose, I suggest to add in your references, two recent papers about this topic: PMID: 34058052, PMID: 34036720

RESPONSE 3 : We are plenty aware of this main limitation. We add consequently this remark in the manuscript. We also integrate the recent published data related to this kind of analysis in our references PMID: 34058052, PMID: 34036720, PMID: 34112706, PMID: 34284044. Humoral response may underestimate and/or overestimate the immunogenicity of vaccine. However, analysis of the humoral response is possible in clinical pratice whereas evaluation of cell-mediated immunity requires a dedicated specific platform which goes beyond the standard of care. Trying to identify accessible tools directly transposable in clinical practive is relevant. According to recently published ROMANOV study (Espi, M. et al, (2021). The ROMANOV study found impaired humoral and cellular immune responses to SARSCov-2 mRNA vaccine in virus unexposed patients receiving maintenance hemodialysis. Kidney Int), humoral response does not guarantee a concomittant cellular response and this discrepancy might be more pronounced in dialysis patients (25% of tested dialysis patients with a humoral response without traces of cellular response). This point also evokes the lack of knowledge about a specific anti-SARS-CoV-2 Spike-Protein-RBD IgG titer which correlates with protection in front of a real-life clinical challenge. These studies are probably in progress. Meanwhile, collecting data on the factors associated with the intensity of the humoral response in particularly vulnerable patients as dialysis patients is mandatory. Finally we should mention that in previously infected patients, there is this “boost-effect” on the intensity of the humoral response (in high responder). This might suggest that there is a link between the IgG levels and the not-analyzed cellular response.

Reviewer #2: Major comments:

1) Why and how was a cut off of 500 AU/ml set to make a decision about additional vaccination? This value seems to me to be quite arbitrary, as absolute values and cut offs differ between commercially available assays. Furthermore, it is still not clear which specific antibody value correlates with protection against (severe) infections. The algorithm and the limitations of this algorithm should be discussed in more detail in the discussion section.

RESPONSE 4 : As discussed aboved in RESPONSE 3, the immunological approach using an antibody response alone to evaluate the immune vaccine response may be too simplistic in comparison to systems-biology approaches. Noteworthy, it missed the evaluation of cellular immune response. However, the well-known correlation between these two systems may support the hypothesis that high IgG titers could reflect a robust coordinated immune response.We made this hypothesis. The cut off of 500 AU/mL was then determined as the IgG titers of the low responder quartitle. Incidentally it represents a ratio of ten (x10) for the positivity threshold of the kit. 

Other population studies should be made to confirm this distribution in other cohorts. Thereby, we have a cutoff to rationalize the access and the distribution to the third vaccine dose which is currently recommanded for all dialysis patients in France. We believe that if a clear seroconversion could be registered (id est with IgG > 500AU/mL), the third dose may be postponed. We thus could prolong the protection conferred by vaccination beyond the reported 8-9 months after the end of complete SARS-CoV-2 vaccination, without endangering our dialysis patients during epidemic waves. Consequently, rare vaccination doses could be better used. The other objective of the algorithm is to support organization and patients circulation in dialysis centers, by identifying the high risk patients who may not have developped enough immune response to be protected and who require maintained vigilance (for example no access to collation during dialysis session, no possibility of dialysis in other centers during epidemic waves, no possibility of intervention without SARS-CoV2 RT-PCR test, reinforced contact precautions, vaccination of relatives). The prospective follow-up of our cohort might demonstrate the safety and efficacy of the proposed algorithm. 

2) On P7 L148-150 something went wrong. Data seems to be missing and the sentence is not completed.

See RESPONSE 2. 

3) I am not convinced that active cancer is really responsible for higher antibody levels after vaccination. Are there any data about BNT162b2 or other vaccines and active cancer? This should be discussed.

RESPONSE 5 : Actually, it was the opposite. An active cancer was defined as a diagnostic of neoplasia without enough delay to permit a theoretical inscription for a transplantation access as mentioned in the legend of table 2). An active cancer has already been associated with a defect in immune response (either due to inflammatory syndrome, hyper catabolic state or effects of oncologic treatment). This information may thus be redundant with the immunocompromised status although only the absence of an active cancer was clearly associated with the high Spike-Protein-RBD IgG anti-SARS-CoV-2 responder group. More precisely there were none active cancers in any of the high responder patients. We add précisions in the text on this point.

4) The authors should also discuss why (based one which data) they stop vaccinating after the second booster dose. E.g. in hepatitis B, a 4th or 5th dose seems to be promising in some patients.

RESPONSE 6 : We thank reviewer 2 for his comment. The hepatitis B virus vaccine nephrologic experience during the last decades indeed brings some tools to organize vaccine allocation. That’s why a third dose was rapidly recommanded in kidney transplant recipients as in dialysis patients in France. However we don’t forget that SARS-CoV-2 vaccines should be considered as a rare and precious therapy. That’s why we proposed to first give this third dose to our non responder patients. On the 11 patients who don’t seroconvert, one died during the follow-up. On the 10 remaining patients, 7 developed a humoral response with Spike-Protein-RBD IgG anti-SARS-CoV-2 reaching the positivity threshold of the kit, although among the 7, only 3 had IgG titers above 500 AU/mL. We modified Figure 1 according to these results. Interestingly, we found an excellent global seroconversion rate of 98.4% in our cohort after a rationalized strategy for the allocation of the third dose. 

Minor comments:

5) "Patients on maintenance dialysis are at risk of developing severe disease from Coronavirus 2019 (COVID-19), in severe acute respiratory syndrome coronavirus 2 (SARS-CoV-2)." This sentence of the introduction is not correct. The last part should be "caused by SARS-CoV-2" or something like that.

RESPONSE 7 : We modified the text as followed : « Patients on maintenance dialysis are at risk of developing severe caused by SARS-CoV-2. »

6) Please use the term "pandemic" instead of "pandemy" throughout the manuscript (e.g. P4 L75).

We corrected the text as suggested.

7) Please delete the following sentence: "Statistical analysis was performed using conventional methods."

We corrected the text as suggested.

8) Please write SARS-CoV-2 and COVID-19 consistently instead of "SARS COV2" or "SARS CoV-2" or "COVID19".

We made the various modifications in the text as suggested.

9) On P6 L131 "seroconversion" is written incorrectly.

We corrected the word.

10) Please write "SARS-CoV-2 vaccination" and not "anti-SARS-CoV-2 vaccination" throughout the manuscript.

We made the various modifications in the text as suggested.

11) Ad Table 1. What does "Figru" under the section "Time after last vaccination injection"?

We corrected Table 1 with the proper result.

12) "In" is misspelled on P10 L188.

We corrected the word.

13) What were the 2 contraindications for vaccination?

RESPONSE 8 : Two patients with an active infection with suspicion of bacterial infection and initiation of antibiotherapy were contraindicated for vaccination.

14) Regarding the algorithm: Under the 3rd dose "IgG > or < 50" is stated. Did you mean 500 as after the second vaccination? What does "Adapted to circulating viral variants" exactly mean?

RESPONSE 9 : In the algorithm we suggest

- Serologic tests should be performed only for the dialysis patients at risk for low response if ressources or techniques are limited.

- The recommanded third for dialysis dose should be allocated in priority for the low responder and as soon as possible when the results is known. If serologic tests are not available, this third dose could be reserved for the patients at risk of low response (for example immunocompromised status or age >80 years old, or no iRAS therapy, or Lymphocytes <0.8G/L, or CRP >20mg/L).

- On the opposite, the third dose should be delayed for the others until their IgG titers drop below 500AU/mL with monitoring every 3-6 months during pandemic times.

- The dialysis patients with no seroconversion (only 3 patients out of the 193 after the complete round of the third vaccinations) should be identified for prespecified precautions as discussed in RESPONSE 4. Considering this very low prevalence, we then suggest to reserve these precautions to the population of low responders (IgG <500AU/mL).

We modified the algorithm in Figure 2 to integrate these considerations and add précisions in the text to support and explain this strategy.

« Adapted to circulating viral variants » anticipates the continuation of the pandemic. It postulates that new generation of mRNA vaccines might be generated to better fit with evolution of the virus. 

Reviewer #3: This review is on the statistical aspects of the paper.

The statistical analysis part is well-written. I only have 2 minor comments.

1. page 5, line 114, why the p-value threshold is set at 0.15?

2. Table 1. the p-value should use the same number of significant digits.

1. Although there is no absolute threshold for p-value in univariate analysis for variables included in multivariate analysis to our knowledge, we modified for a more classical p-value of 0.1

2. We standardized the p-value for the same number of digits in table 1.

6. PLOS authors have the option to publish the peer review history of their article (what does this mean?). If published, this will include your full peer review and any attached files.

Do you want your identity to be public for this peer review? For information about this choice, including consent withdrawal, please see our Privacy Policy.

Reviewer #1: No

Reviewer #2: No

Reviewer #3: No

---

## [Editor Report · Decision Letter 1]

7 Sep 2021

Humoral response after SARS-CoV-2 mRNA vaccines in dialysis patients: Integrating anti-SARS-CoV-2 Spike-Protein-RBD antibody monitoring to manage dialysis centers in pandemic times.

PONE-D-21-16866R1

Dear Dr. BACHELET,

We’re pleased to inform you that your manuscript has been judged scientifically suitable for publication and will be formally accepted for publication once it meets all outstanding technical requirements.

Kind regards,

Etsuro Ito

Academic Editor

PLOS ONE

---

## [Editor Report · Acceptance letter]

24 Sep 2021

PONE-D-21-16866R1 

Humoral response after SARS-CoV-2 mRNA vaccines in dialysis patients: Integrating anti-SARS-CoV-2 Spike-Protein-RBD antibody monitoring to manage dialysis centers in pandemic times. 

Dear Dr. BACHELET:

I'm pleased to inform you that your manuscript has been deemed suitable for publication in PLOS ONE. Congratulations! Your manuscript is now with our production department. 

Kind regards, 

on behalf of

Prof. Etsuro Ito 

Academic Editor

PLOS ONE